# The Potential Neuroprotective Effects of Extracts from Oat Seedlings against Alzheimer’s Disease

**DOI:** 10.3390/nu14194103

**Published:** 2022-10-02

**Authors:** Won Seok Lee, Hae-June Lee, Ji Yeong Yang, Hye-Lim Shin, Sik-Won Choi, Jong-Ki Kim, Woo Duck Seo, Eun Ho Kim

**Affiliations:** 1Department of Biochemistry, School of Medicine, Daegu Catholic University, Nam-gu, Daegu 42472, Korea; 2Division of Radiation Biomedical Research, Korea Institute of Radiological and Medical Sciences, Seoul 01812, Korea; 3Division of Crop Foundation, National Institute of Crop Science, Rural Development Administration, Jellabuk-do, Deokjin-gu, Jeonju 55365, Korea; 4Forest Biomaterials Research Center, National Institute of Forest Science (NIFoS), Korea Forest Service (KFS), Jinju 52817, Korea; 5Department of Biomedical Engineering & Radiology, School of Medicine, Daegu Catholic University, Daegu 42472, Korea

**Keywords:** OSE, Alzheimer’s disease, β-amyloid, BACE1

## Abstract

The physiological or dietary advantages of germinated grains have been the subject of numerous discussions over the past decade. Around 23 million tons of oats are consumed globally, making up a sizeable portion of the global grain market. Oat seedlings contain more protein, beta-glucan, free amino acids, and phenolic compounds than seeds. The progressive neurodegenerative disorder of Alzheimer’s is accompanied by worsening memory and cognitive function. A key indicator of this disorder is the unusual buildup of amyloid-beta protein (or Aβ) in human brains. In this context, oat seedling extract (OSE) has been identified as a new therapeutic candidate for AD, due to its antioxidant activity and AD-specific mechanism of action. This study directly investigated how OSE affected AD and its impacts by examining the cognitive function and exploring the inflammatory response mechanism. The dried oat seedlings were grounded finely with a grinder, inserted with 50% fermented ethanol 10 times (*w*/*v*), and extracted by stirring for 10 h at 45 °C. After filtering the extract by 0.22 um filter, some of it was used for UHPLC analysis. The results indicated that the treatment with OSE protects against Aβ25–35-induced cytotoxicity in BV2 cells. Tg-5Xfad AD mice had strong deposition of Aβ throughout their brains, while WT mice did not exhibit any such deposition within their brains. A drastic reduction was observed in terms of numbers, as well as the size, of Aβ plaques within Tg-5Xfad AD mice exposed to OSE. This study indicated OSE’s neuroprotective impacts against neurodegeneration, synaptic dysfunction, and neuroinflammation induced by amyloid-beta. Our results suggest that OSE acts as a neuroprotective agent to combat AD-specific apoptotic cell death, neuroinflammation, amyloid-beta accumulation, as well as synaptic dysfunction in AD mice’s brains. Furthermore, the study indicated that OSE treatment affects JNK/ERK/p38 MAPK signaling, with considerable inhibition in p-JNK, p-p38, and p-ERK levels seen in the brain of OSE-treated Tg-5Xfad AD mice.

## 1. Introduction

Oats account for much of the global grain market, and worldwide consumption is around 23 million tons [1]. Oats do not contain gluten, so they are useful for people with celiac disease and are recognized as a portion of food with many health benefits, compared to other grains [2]. Besides basic primary metabolites, oats contain micronutrients, beta-glucans, and various phenolic compounds, which have many health benefits, such as antioxidant activity [3,4,5,6]. In the last 10 years, there has been a lot of discussion about the nutritional or physiological benefits of germinated grains [7,8]. Oat seedlings contain more protein, beta-glucan, free amino acids, and phenolic compounds than seeds [9]. In addition, various secondary metabolites, including avenacoside, have been identified in sprouted oats, and various physiological activities have been found [10]. Secondary metabolites of oat seedlings have various physiologically active functions, such as osteoblast differentiation [11], the anthelmintic effect [12], and improvement in acne [13].

Alzheimer’s disease (AD) is one of the most prevalent neurodegenerative disorders and is attributed to several factors, including the emergence of amyloid-(A) plaques, hyperphosphorylation, microglia activation, and neuronal degeneration [12,13]. Progressive memory loss and cognitive impairment are common in AD patients. None of the available treatments for AD can delay or halt the progression of the disease; they can only treat symptoms. The abnormal accumulation of Aβ is still the key event in the onset and progression of AD, although there are many contributing factors [14,15,16]. The sequential cleavage of APP by β-secretases like BACE1 and γ-secretase complex produces Aβ. There are two main species, Aβ40 and Aβ42, with Aβ42 constituting the majority of Aβ plaques in the brain of AD patients [17,18,19,20]. Although postulated, the A mechanism of neurotoxicity has not yet been evidenced. Learning and memory deficits can be better understood by using an animal model of AD that exhibits Aβ pathology. The role of inflammation in AD is a more recent area of interest. Research utilizing animal models has shown that inflammation can impair cognitive function [21], as well as cause neuronal damage and synaptic loss in vivo and in vitro [22,23,24]. Although inflammation and microglia activation are thought to have a neuroprotective effect in short-term situations, over time, they may cause neurotoxicity and an increase in Aβ load [21,25]. The activation of microglia by Aβ is thought to draw them to plaques and optimize phagocytosis [26]. Potentially, the microglial response to Aβ is secure, but after prolonged activation, the microglia take on a negative role and cause a degradation feed-forward loop [27].

Pharmaceutical treatment using natural products is one of the oldest methods that has continued in the past human history [14]. The demand for natural products is rising as concerns about the effects of chemicals on human health and the environment increase. Interestingly, there is also a therapeutic agent for AD derived from natural products. Galantamine, isolated from the bulbs of *Galanthus nivalis*, is the only drug with plant origin recognized for AD [15]. Since then, plant-based natural products have been continuously studied as AD therapeutic agents, and seven plant-derived compounds have been identified as anti-Alzheimer drug candidates and AD targets [16]. We confirmed that oat seedlings possess a variety of secondary metabolites, and among them, avenacoside-type flavonoids are the major compounds. Avenacoside A (AVN-A) and B were identified as oat seedlings [10]. We identified OSE with antioxidant activity and an AD-specific mechanism of action and confirmed that OSE can be used as a new therapeutic candidate. Sprouting involves various metabolic/biochemical/physiological processes that release active nutrients to the tissues of growing plants [9,10], often offering many health benefits [9,11]. For this reason, it is a potent means of improving nutritional profiles [11]. Sprouting seeds, on account of their nutritional value, have lately elicited a lot of interest from all over the world, with researchers attempting to determine the availability of new ingredients under different circumstances [9,10]. This study aimed to provide experimental evidence of how OSE affected AD by examining cognitive function and exploring the inflammatory response mechanisms responsible for exerting these impacts.

## 2. Materials and Methods

### 2.1. Chemicals and Antibodies

The information about chemicals and antibodies used in the method is as follows: charged aerosol detector (CAD) (Thermo Scientific, Waltham, MA, USA), 0.1% fomic acid (Sigma Aldrich, Saint Louis, MO, USA), acetonitrile (Sigma Aldrich), amyloid-β (Santa Cruz, Dallas, TX, USA, sc-53822, 1:500), BACE1 (Invitrogen, Waltham, MA, USA, PA5-19952, 10 μg/mL), APP (Invitrogen, 14-9749-82, 2.5 μg/mL), Iba-1 (Abcam, Cambridge, UK, ab178846, 1:2000), GFAP (BD Biosciences, Dickinson, ND, USA, BD-556328, 5 μg/mL), β-actin (Santa Cruz, sc-8432, 1:1000), phospho-Tau (Invitrogen, 44-752G, 1:1000), Bace1 (Invitrogen, PA5-19952, 1:1000), phospho-JNK (Santa Cruz, sc-6254, 1:1000), phospho-ERK (Santa Cruz, sc-7383, 1:1000), phospho-p38 (Cell Signaling, Danvers, MA, USA, csD3F9, 1:1000), anti-goat mouse (Enzo Life Sciences, Farmingdale, NY, USA, ADI-SAB-100-J, 1:2000), and anti-goat rabbit (Enzo Life Sciences, ADI-SAB-300-J, 1:2000).

### 2.2. Plant Materials, Extract Chemical Isolation Method and Ultra-High Performance Liquid Chromatography (UHPLC) Analysis

In 2020, oats were planted in artificial soil within a growth chamber purchased from Jeonbuk, Korea-based National Institute of Crop Science (NICS), Rural Development Administration (RDA). After being soaked inside water for one day at 20 °C, the oats were taken out and germinated for two days in a dark state, with 65% humidity. Seedlings were harvested on the 8th day, washed once with sterile deionized water, and then freeze-dried at −78 °C. The dried oat seedlings were grounded finely with a grinder, put into circulation with aqueous fermented ethanol (50%), and extracted by stirring for 10 h at a 45 °C temperature [28]. After filtering and centrifugation, the supernatant was concentrated by spray drying. After filtering the extract with a 0.22 μm filter, some of it was used for UHPLC analysis, and the remaining extract was concentrated using a rotary evaporator (EYELA, Tokyo, Japan) to make OSE. Analysis of the extract was performed by UHPLC Dionex Ultimate 3000, with CAD used for analysis. For chromatogram separation, an Acclaim PolarAdvantage II C18 column 5 μm 4.6 × 250 mm (Thermo Scientific) was used, the oven temperature was 35 °C, and the mobile phase was 0.1% formic acid in water (A) and acetonitrile (B). Flow rate was 1 mL/min, and the mobile phase concentration gradient is as follows: 0–3 min, 15% B; 3–9 min, 15–20% B; 9–16 min, 20–22% B; 16–20 min, 22–30% B; 20–36 min, 30–45% B; 36–38 min, 45–90% B; 38–40 min, 90% B; 40–42 min, 90–15% B; and 42–50 min, 90–15% B. The injection volume was 1.3 µL, and the CAD power function was 1.3.

### 2.3. Cell Culture

As far as AD is concerned, HT22 cells are known for their utilization with in vitro cellular models, and their functional cholinergic attributes are associated with AD’s cognitive ailments. The growth of this cell line took place within a Dulbecco’s Modified Eagle Medium (DMEM) that comprised of 1% penicillin-streptomycin solution, as well as 10% fetal bovine serum. These cells, growing at an exponential pace, were kept inside an incubator at 37 °C (5% CO_2_ and air and 95% air).

BV2 cell culture and treatment of cells: BV2 microglial cells were maintained in DMEM containing 10% heat-inactivated FBS and 1% antibiotic-antimitotic agents at 37 °C in a humidified incubator, with 95% O_2_ and 5% CO_2_, as previously described [29]. Cells were treated with LPS (1 µg/mL) and/or a series of concentrations of OSE for 36 h. Cells exposed to serum-free media containing 0.3% DMSO were used as the vehicle-treated control groups.

### 2.4. Cell Viability Assay

For the incubation of 2 × 10^4^ cells in a 96-well plate for 24 h, the CCK-8 kit (Dojindo, Rockville, MD, USA, CK04) was treated for 2 h [30], and the cell viability was measured by reading at 450 nm using a plate reader (Hangzhou Allsheng Instruments, Hangzhou, China, AMR-100).

### 2.5. Cell Death Detection Assay

For the incubation of 1 × 10^5^ cells in a 6-well plate for 72 h, the cell death detection was proceeded using a cell death detection ELISA kit (Roche, Basel, Switzerland, China, 1154467500) according to the manufacturer’s instructions [31]. Cell death detection was measured by reading at 405 nm using a plate reader (Allsheng, AMR-100).

### 2.6. Fluorescent Measurement of Intracellular Reactive Oxygen Species (ROS)

The fluorescent probe 2′,7′-dichlorofluorescin diacetate (DCFH-DA) was utilized for the analysis of intracellular ROS. For fluorocytometrical estimation, cells were treated with OSE for two days and were maintained at room temperature for half an hour, with 10 μM of DCFH-DA in 5 mL PBS. Fluorescence was measured with a flow cytometer (Becton Dickinson, Franklin Lakes, NJ, USA) in accordance with the manufacturer’s protocols [32].

### 2.7. Animals

The 5xFAD[B6SJL-Tg(APPSwFlLon,PSEN1*M146L*L286V)6799Vas/Mmjax] transgenic mice were purchased from the Jackson Laboratory (Bar Harbor, ME, USA). Female WT, as well as Tg-5xFAD AD mice, were used and allocated into various groups. Exposure to OSE began when the animals were 1.5 months old, which means that their brains were yet to mature. This was because the development of intraneuronal Aβ aggregation took place genetically within Tg-5XfaD AD mice from this point onward. Experiments were carried out in compliance with approval from the Institutional Animal Care and Use Committee (DCIAFCR-210629-11-YR). All institutional and international norms for animal care were complied with. Following 1 week of adaption, all Tg-5xFAD mice were randomly divided into either the AD model group or the OSE + AD model group (*n* = 5 in each group). Mice were also randomly divided into either the normal control (NC) group or the OSE + NC group (*n* = 5 in each group). The mice in the OSE + NC and OSE + AD model groups were administered OSE (100 mg/kg) orally once daily for 6 weeks. Following treatment, behavioral tests and biochemical experiments were performed.

### 2.8. Object Recognition Test (ORT)

Overnight and before the training, the mice were placed in a test chamber under the supervision of a twelve-hour light–dark cycle at 23 ± 1 °C and 50–60% humidity, with food/water ad libitum. During the course of training, two round filter units whose height and diameter was the same (27 mm and 33 mm, respectively) were positioned inside the chamber. Then, the mice were allowed to carry out an exploration for ten minutes. After one day, one object was substituted using a plastic cone, whose height was 30 mm with its diameter being 25 mm. The definition of object recognition was made based on the time put in to touch or sniff the new object within five minutes [33]. EthoVision XT8.5 was used to record and analyze the training and assess the trials [34].

### 2.9. Radial Arm Maze Test (RAM)

A neurocognitive RAM test was performed before and after OSE treatment on mice groups, including the untreated and non-Tg wild type. The spatial working memory was tested using an eight-arm radial maze as described [35]. One reward cup was placed atop a platform at each arm’s distal end. Each mouse first underwent 10-min habituation training trials on the radial arm maze over 3 consecutive days. After acclimation, the mouse was allowed to access open arms throughout the testing (two arms were baited, separated by 135°), and the baited arms were searched. The trial was terminated when the animal located the baited arms and consumed the food reward. The number of arms that the mouse visited before visiting the two baited arms, including revisits, was counted. A correct choice was counted only if the mouse approached and ate from a baited food cup. A visiting error was considered a spatial working memory error and occurred when a rat re-entered an arm that was unbaited or had been previously baited [36]. These performance measures were acquired before and after OSE treatment and analyzed by analysis of variance (ANOVA).

### 2.10. Tissue Preparation

The sacrifice of mice was made for harvesting tissue of the brain after performing behavioral tests across two months following treatment with OSE. As far as histological analysis is concerned, three mice’s (in each group) left hemispheres were placed in a 4% paraformaldehyde solution, whereas the right hemispheres were placed in RNA later^®^ at 4 °C for one night. The dissection of the hippocampus took place in the brain before being placed at −80 °C for PCR analysis and the microarray. The hippocampus and entorhinal cortex, according to the atlas of Paxinos and Franklin [37], of three mice from each group were dissected for Western blotting analysis.

### 2.11. Immunohistochemistry

The sections’ immunostaining was carried out using a Vectastain Elite ABC kit purchased from US-based Vector Laboratories Inc. To retrieve the antigen, the first step entailed placing these sections inside citrate buffer before being boiled inside the water for half an hour. In terms of immunoperoxidase labeling, 0.3% H_2_O_2_ was used to block endogenous peroxidase in methanol at room temperature for around fifteen minutes. Regarding immunohistochemistry, the blocking of sections was done using horse serum before being incubated at 4 °C for one night, with an anti-mouse BACE1, GFAP, Iba-1, or human Aβ antibody. After being incubated with mouse IgG or biotinylated goat anti-human at RT for 30 min, this was followed by immunoreaction with a peroxidase complex avidin-biotin-based) at RT for half an hour. The DAB kit was used to develop the peroxidase reaction. In all experiments, it was seen that the primary antibody would be omitted for some sections and counterstained using the hematoxylin of Harris before being mounted. GFAP and Iba-1 positive cell counting were quantified by using the Image J program (1.8.0 version, NIH, Bethesda, MD, USA).

### 2.12. Analysis of Western Blot

Total proteins from cells were extracted in RIPA buffer (50 mM Tris-Cl, pH 7.4; 1% NP-40; 150 mM NaCl, and 1 mM EDTA) before being supplemented with protease inhibitors (1 mM PMSF, 1 μg/mL aprotinin, 1 μg/mL leupeptin, and 1 mM Na_3_VO_4_) and then quantified using the Bradford method. Protein samples (30 μg) were separated by SDS/polyacrylamide gel electrophoresis and transferred to a nitrocellulose membrane, used as described previously [38].

### 2.13. Real-Time Polymerase Chain Reaction (qRT-PCR)

RNA from mouse brain regions was purified using RNeasy Mini Kit (Qiagen, Hilden, Germany). APP cDNA was synthesized using the following primers. Sense: 5′-TGCTGGCAGAACCCCAGATCG-3′, Antisense: 5′-TTCTGGATGGTCACTGGCTGG-3′. The gene expression level was normalized to the housekeeping gene β-actin, and the following primer sequences were used. Sense: 5′-TGCTTCTAGGCGGACTGTTACTGA-3′, Antisense: 5′-TCGCCTTCACCGTTCCAGTTTT-3′. QRT-PCR assay was performed in duplicate on cDNA samples proceeded according to the references [39,40]. Fold change mean value was analyzed with a statistical test using Prism.

### 2.14. Statistical Analysis

Statistical significance was determined using Tukey’s method with one-way ANOVA. Differences were deemed statistically important in case the *p* value was lower than 0.001 or 0.05. (* *p* < 0.05; ** *p* < 0.01; *** *p* < 0.001).

## 3. Results

### 3.1. Effect of OSE on Cell Viability and Treatment with OSE Protects against LPS-Induced Cytotoxicity in BV2 Cells

To test the effects of OSE and compound AVN-A of OSE on neuroinflammation, we first examined whether OSE and AVN-A are toxic to BV2 microglial cells. Cells were treated with vehicle (1% DMSO), OSE (25, 50, 100, or 200 μM), or AVN-A (10, 20, or 40 μM) for 24 h, and we found that OSE and AVN-A did not induce microglial cell toxicity at concentrations up to 100 μM and 20 μM (Figure 1A). When BV2 cells were pretreated for 2 h with OSE or AVN-A, LPS (2 μM)-induced cytotoxicity was significantly reduced (Figure 1B,C). To investigate the effect of OSE and AVN-A on cell death and ROS production, the levels of ROS in BV2 cells were detected using a ROS detection kit. Pre-incubation with OSE and AVN-A significantly diminished the levels of ROS in LPS-stimulated cells (Figure 1D). These results demonstrated that the LPS-induced loss of cell viability was partially attenuated by OSE and AVN-A; thus, they may inhibit LPS-induced production of ROS in microglia.

### 3.2. OSE and AVN-A Exposure Effectively Inhibits Brain Aβ Deposition and Protein Expression

According to the ELISA assay, HT-22 cells exposed to OSE and AVN-A have significant reductions in Aβ 42 and Aβ 40 (Figure 2A). Moreover, exposure to OSE lowered the Aβ42/Aβ40 ratio in HT-22 cells treated with OSE (Figure 2A). The Aβ42/Aβ40 species’ relative levels and distinct ratios played an important role in the pathogenesis of AD. Notably, immunohistochemistry assessment for Aβ across sections of the brain from Tg-5xFAD AD mice and WT, after exposure to OSE, emphasized the CA1 of the entorhinal cortex and hippocampus—the primary lesion within the realm of AD pathology. According to the analysis, Tg-5XFAD AD mice witnessed a very strong deposition of Aβ throughout their brains, while WT mice did not exhibit any such deposition within their brains. A drastic reduction was observed in terms of numbers, as well as the size of Aβ plaques within Tg-5XFAD AD mice exposed to OSE (Figure 2B). It was confirmed by the Western blot that Aβ decreased when OSE and AVN-A were administered to both cell lines (Figure 2C,D) and Tg-5xFAD AD mice (Figure 2E,F). The findings were in line with the Aβ findings [14,15,16].

### 3.3. BACE1 Expression in the Brain after OSE and AVN-A Exposure

Western blotting and RT-PCR were done on brain tissue and was done to determine whether APP processing and Aβ protein degradation were affected by OSE exposure. No dissimilarities were found within the APP as far as wild-type mice were concerned. APP levels rose in terms of the entorhinal cortex and hippocampus. At the same time, APP was observed in the brain of Tg-5xFAD AD mice. Following OSE and AVN-A exposure, the level of APP declined in the HT22 cells and Tg-5xFAD AD mice brains (Figure 3A,B,D,E), and the expression level of mRNA was correlated (Figure 3C). BACE1 immunostaining was done across brain sections, after which elevated BACE1 expression patterns were seen in the aforementioned regions of Tg-5xFAD AD mice, compared to the WT mice (Figure 3A,B,D,E). Having said that, Tg-5XFAD AD mice exposed to OSE exhibited a significant drop in BACE1 expression across the brain section, in comparison with control Tg-5xFAD AD mice. As far as WT mice are concerned, there was a minor BACE1 induction caused by exposure to OSE, which was in contrast to the Tg-5xFAD AD mice results (Figure 3F).

### 3.4. OSE and AVN-A Exposure Decreases Neuroinflammatory Cells

An elevated GFAP expression was observed in Tg-5xFAD AD mice when compared with WT mice. At the same time, strong GFAP positive signaling was seen in Tg-5xFAD AD mice as well. The heightened GFAP was validated by Western blotting brain homogenates across various regions of the brain. Nevertheless, the entorhinal cortex, as well as the hippocampus of the brain section and HT22 cells exposed to OSE and AVN-A, exhibited a significantly lower expression of GFAP (Figure 4A–E). In terms of WT mice, exposure to OSE indicated no or minimal impact on the expression of GFAP. Notably, immunoblotting data for Iba1 also exhibited similar impacts; a lower expression of Iba1 was found in Tg-5xFADAD mice exposed to OSE compared to Tg-5xFAD AD mice. However, in WT mice, the Iba1 expression did not change following exposure to OSE (Figure 4A–E).

### 3.5. OSE Exposure Ameliorates Memory Impairment

The next step entailed determining whether exposure to OSE impacted the AD mice’s memory functions. The entire experimental scheme of this study is summarized in Figure 5. To assess the correlation of OSE treatment with the improvements in cognitive function, relatively aged AD mice (aged 5–12 months), untreated and OSE treated, and non-Tg WT (*n* = 5) or treated relatively early onset (aged 1–4 months) AD mice were subjected to two spatial memory tests, the RAM and ORT tests, and the results were compared (Figure 6A–D). Exposure to OSE substantially mitigated the Tg-5xFAD AD mice’s inhibited spatial memory. However, no OSE exposure benefit was imparted to wild-type animals.

### 3.6. Mechanism of OSE and AVN-A Effects on AD-Related Phenotypes

The current study examined whether exposure to OSE and AVN-A impacts MAPK signaling. According to the Western blot analysis, phosphorylated (p) c-Jun N-terminal kinase (JNK), p-extracellular signal-regulated kinases (ERK), and p-p38 mitogen-activated protein kinase (p38 MAPK) levels declined considerably in LPS + OSE treatment when compared with LPS. Thus, this result indicated the potential contribution of JNK/ERK/p38 MAPK signaling within OSE administration (Figure 7A).

## 4. Discussion

Only symptoms of AD are currently being treated. Presently, the FDA has only approved four medications for AD: three AChE inhibitors (donepezil, galantamine, and rivastigmine), and one NMDA antagonist (memantine) [41]. Tacrine, a fifth drug, and another AChE inhibitor were taken off the market because it was hepatotoxic [42]. However, the benefits of these medications in treating the symptoms are only marginal. Additionally, they are ineffective at preventing the loss of neurons, brain atrophy, and ensuing gradual decline in cognition [41]. In the recent past, natural products have elicited importance for developing treatment options for CNS ailments owing to their neuroprotective impacts [43], especially due to their anti-inflammatory, immunomodulatory, and antioxidant capacities. As per the findings of this study, avenacoside delayed the effects of Aβ- on memory and learning, suggesting that OSE may be effective in treating neurodegenerative disease by preventing the onset of the condition. In a previous article, we discussed the phytochemical profile of OSE and UPLC-CAD chromatograms of its polyphenol content [28]. Inhibitory effects of all compounds on bacterial neuraminidase were reported in a recent study, and as a result, OSE may also contain significant and valuable resources for promising therapeutic or precautionary agents against infectious diseases. To the best of our knowledge from the viewpoint of AD, this study also shows for the first time that OSE can attenuate microglial activation induced by LPS, as well as pro-inflammatory responses within BV2 microglial cells. OSE is also known to curtail the production of TNF-α, IL-1β, and NO, as well as ROS, which can be mediated via MAPK signaling pathway inhibition in BV2 cells. Therefore, as a natural product, OSE has considerable potential in drug discovery against AD, with a strong emphasis on clinical and safety aspects. This study showed OSE’s neuroprotective impact against neurodegeneration, synaptic dysfunction, and neuroinflammation induced by amyloid-beta. OSE effectively suppressed particular markers of activated astrocytes and microglia, including GFAP and Iba-1. In addition, the expression of p-NF-kB and TNF-α, as well as IL-1β, were also evaluated in the brains of mice treated with Aβ_1–42_ + OSE, after which the expression of amyloidogenesis-related markers and apoptotic markers, as well as synaptic proteins, were investigated. Our results suggested that OSE does act as a neuroprotective agent to combat AD-specific apoptotic cell death, neuroinflammation, amyloid-beta accumulation, and synaptic dysfunction in AD mice’s brains. Furthermore, it was also found that OSE treatment affects ERK/p38 MAPK signaling, with considerable growth in p-p38 p-ERK levels seen in the brain of APP/PS1 mice treated with OSE. In a study by Colombo et al. [44], JNK was found to regulate the phosphorylation as well as degradation of the amyloid precursor protein (APP) in an AD model, which led to significant reductions in Aβ fragments, Aβ oligomers, and APP, indicative of a linkage between the APP metabolism and JNK pathway. Contrastingly, this study observed that p-JNK levels did not change in APP/PS1 mice treated with OSE, relative to AD and WT mice. It was also found that the ERK/p38 MAPK pathway can regulate neuronal differentiation and neuroinflammation, as well as synaptic plasticity [45,46,47]. Moreover, p38 MAPK is regarded as a new therapeutic target in AD treatment [48]. According to Khan and Alkon (who showed Alzheimer’s disease-specific changes in the Erk1/Erk2 phosphorylation ratio), AD-specific changes were found in the p-Erk1/Erk2 ratio. In this study, OSE administered orally in APP/PS1 mice was found to suppress phenotypes related to AD by regulating the pathway of ERK/p38 MAPK signaling. Natural products have long been used for medicinal purposes. The endeavor to develop natural products in the form of possible therapies, especially via isolation and extraction techniques, contributed to the creation of 63% of drugs derived from natural products from 1981 to 2006 [49,50]. A large part of these efforts was dedicated to developing natural anti-AD agents [43,51] due to their structurally diverse attributes, as well as biological activities [52]. As of now, the FDA has only approved some AChE inhibitors and NMDA receptor antagonists for AD treatment, despite the continued research efforts. Thus, the current paper discussed the potential use of OSE natural products as well as their molecular targets as potential anti-AD agents. Although natural products, along with their isolated natural compounds, have gained recognition as neuroprotective agents in treating AD, a large number of them are still to be tested, and it is difficult to observe their clinical efficacy due to a myriad of factors [53]. To begin with, the natural source materials for natural products are predicated on genetic factors, as well as a host of extraneous components, such as harvest time, environmental conditions, and agricultural and collection practices for source materials, thus rendering quality control difficult over the raw materials. The methods and requirements related to quality control of the eventual combination of natural products comprised of hundreds of natural constituents are even more complicated. Due to its well-balanced multitarget profile, the compound could be seen as a useful starting point for future hit-to-lead efforts in discovering new candidates to treat neurodegenerative diseases. The overall goal of this study was to investigate whether OSE could provide protective effects against AD, and if positive, could be useful for future AD drug development. The findings of this study should also facilitate the creation of suitable cultivars that may one day yield foods with potential health benefits. These findings imply that OSE may play a critical role as a clinical treatment for neurodegenerative disorders like Parkinson’s disease, Alzheimer’s disease, and senile dementia. However, it is necessary to conduct further in vivo investigations using animal models to ascertain the long-term efficacy of OSE effectiveness for CNS ailments.

## Figures and Tables

**Figure 1 nutrients-14-04103-f001:**
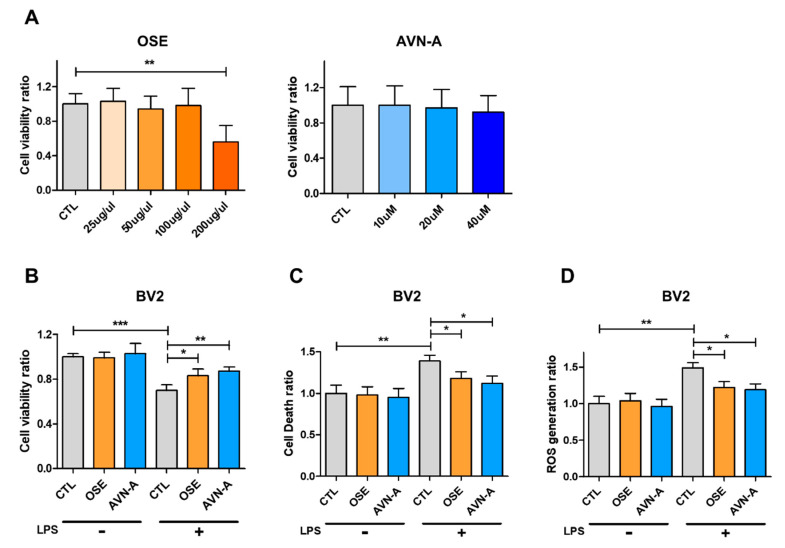
Cytotoxic effects of OSE and AVN-A of OSE extract on BV2 microglial cells. (**A**) Cytotoxicity of OSE and AVN-A in BV2 cells was assessed by MTT assay. (**B**) Following treatment with lipopolysaccharide (LPS), OSE, or AVN-A for 36 h, the viability of the BV2 cells was assessed by MTT assay. Values correspond to the means of three independent experiments and are shown as the percentage of viable cells compared with the viability of untreated cells. * *p* < 0.05, ** *p* < 0.01, *** *p* < 0.001. Analysis of cell death and ROS generation was done by (**C**) cell death detection kit and (**D**) ROS detection kit.

**Figure 2 nutrients-14-04103-f002:**
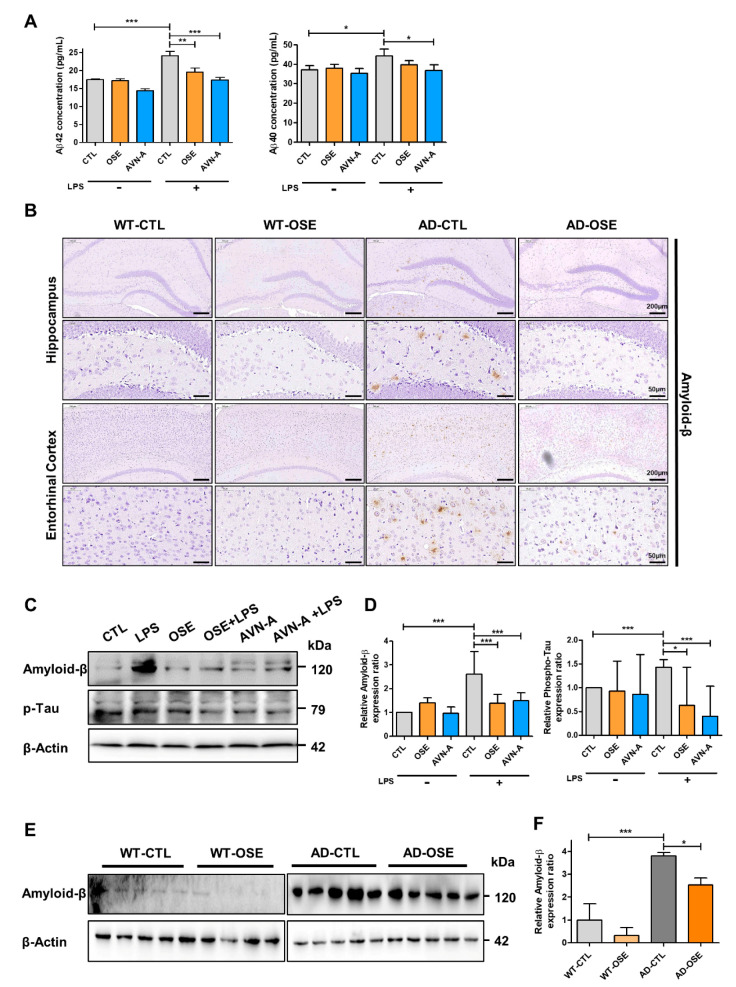
OSE and AVN-A exposure lowers the deposition of Aβ in Tg-5xFAD AD mice. (**A**) The Aβ42 and Aβ40 in the cell lysate were measured by color metric ELISA assay. (**B**) Representative images of Aβ deposition in the CA1 region of the hippocampus and entorhinal cortex of Tg-5xFAD AD and WT mice. (**C**,**E**) Cell lysates (30 µg) of each treated cell line and in vivo were immunoblotted (IB) with indicated antibodies. (**D**,**F**) Validation of differentially expressed proteins were done by Western-blot analysis. * *p* < 0.05, ** *p* < 0.01, *** *p* < 0.001.

**Figure 3 nutrients-14-04103-f003:**
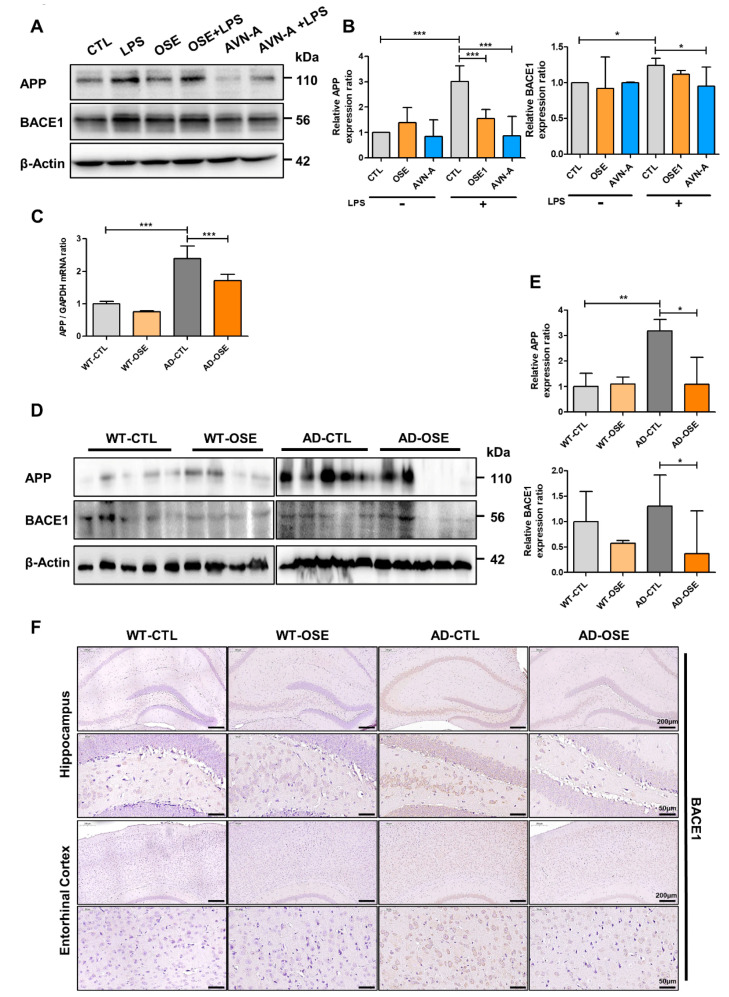
Effects of OSE in different expressions of APP and BACE1 in WT and Tg-5xFAD AD mice brains. (**A**) Western blotting in the HT22 cell line after exposure to OSE and AVN-A. (**B**) Validation of differentially expressed proteins by Western-blot analysis. (**C**) RT-PCR for APP performed using whole brain tissues from WT and Tg-5xFAD AD mice with or without OSE exposure. (**D**) BACE1 Western blotting in the cortex extract, as well as hippocampus, within both WT and Tg-5xFAD AD mice after exposure to OSE. (**E**) Validation of differentially expressed proteins by Western-blot analysis. (**F**) Representative images of BACE1 in the CA1 region of the hippocampus and entorhinal cortex of Tg-5xFAD AD and WT mice. * *p* < 0.05, ** *p* < 0.01, *** *p* < 0.001.

**Figure 4 nutrients-14-04103-f004:**
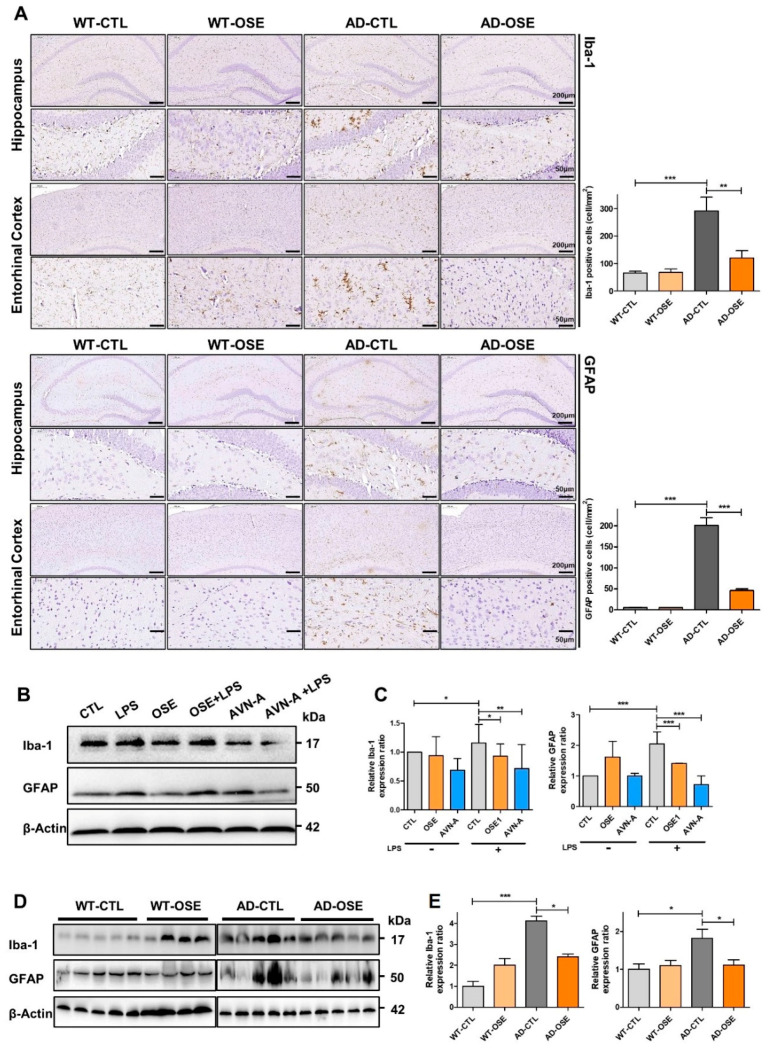
Reduced reactive astrocytes and active microglia. (**A**) Immunohistochemistrical staining of GFAP and Iba1 within the brain sections’ hippocampus and entorhinal cortex in Tg-5xFAD AD and WT mice after exposure to OSE. (**B**,**D**) GFAP and Iba1 Western blotting in cell line and Tg-5xFAD AD mice after exposure to OSE and AVN-A. Actin is being utilized as the loading control. (**C**,**E**) Validation of differentially expressed proteins by Western-blot analysis. * *p* < 0.05, ** *p* < 0.01, *** *p* < 0.001.

**Figure 5 nutrients-14-04103-f005:**
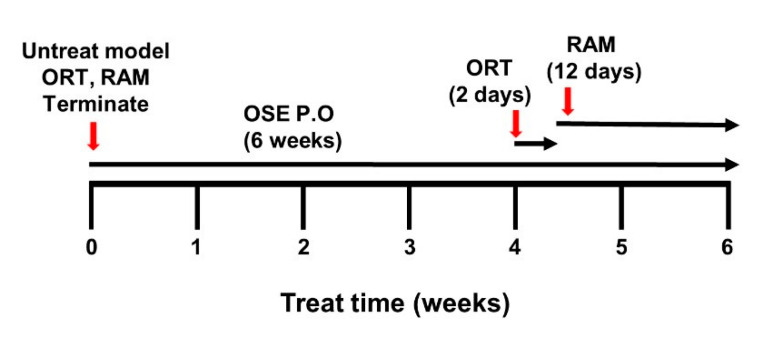
A schematic summary. A schematic summary of the experimental procedure. Total distance covered, as well as the time taken by mice, are in the center zone.

**Figure 6 nutrients-14-04103-f006:**
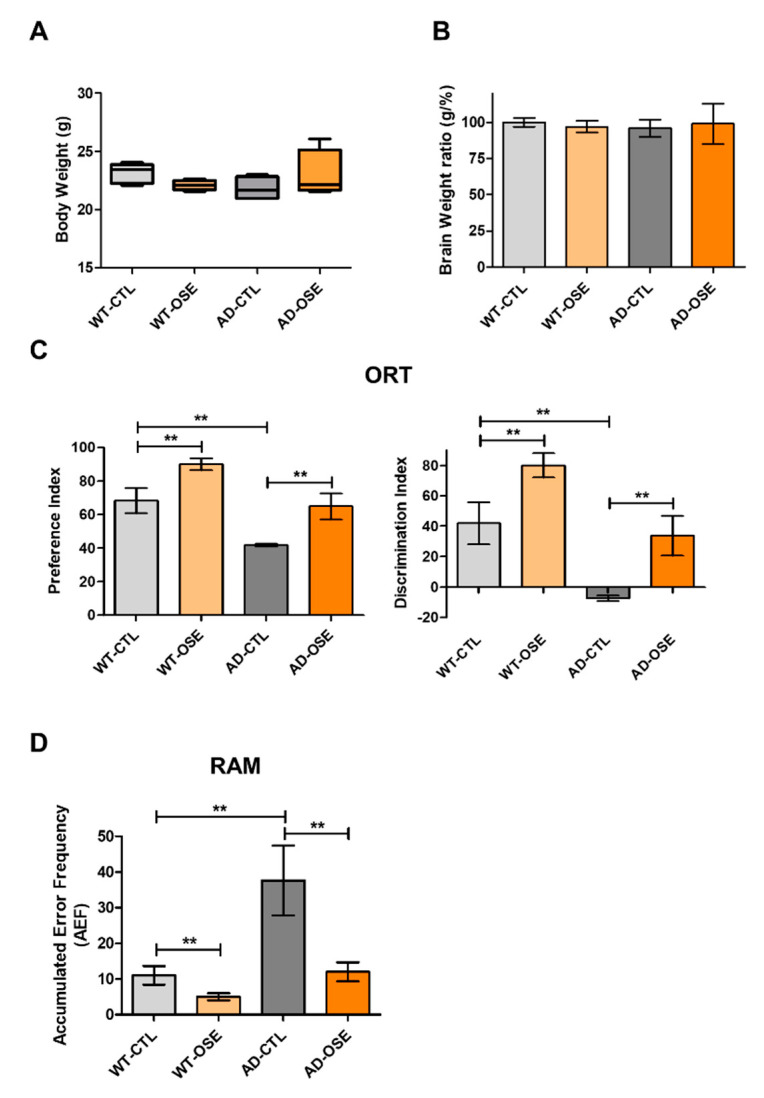
OSE attenuating memory impairment. (**A****,B**) The body and brain tissues of mice were weighed at the last experiment (six weeks). (**C**) Novel object recognition task in Tg-5xFAD AD mice and their control group (non-Tg mice). The discrimination index was calculated as the percentage ratio of *TB*/(*TA* + *TB*) × 100. *TA*: familiar object. *TB*: novel object. (**D**) Radial arm maze test assessed spatial memory. Change denotes the percentage of non-overlapped frequency entries compared with the overall entries within three arms. ** *p* < 0.01.

**Figure 7 nutrients-14-04103-f007:**
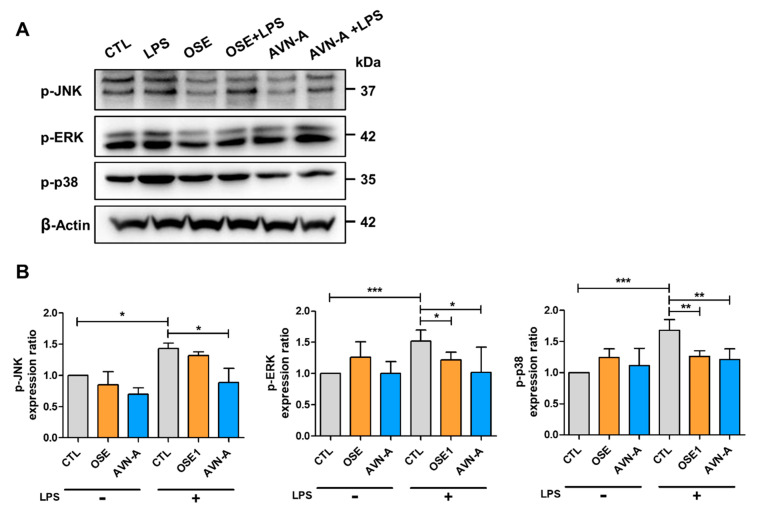
Inhibitory effects of OSE and AVN-A on the phosphorylation of c-JNK, ERK, and p38. (**A**) The expression levels of p-JNK, p-ERK, and p-p38 were measured by Western-blot analysis. (**B**) Validation of differentially expressed proteins by Western-blot analysis. * *p* < 0.05, ** *p* < 0.01, *** *p* < 0.001.

## Data Availability

Not applicable.

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
