# Peer review of "The Potential Neuroprotective Effects of Extracts from Oat Seedlings against Alzheimer’s Disease"

_nutrients, 2022, doi:10.3390/nu14194103_

Round 1
Reviewer 1 Report
In this Manuscript, the Authors would investigate the effects of OSE-1, a natural compound, in ameliorating AD pathology, using an in vitro and an in vivo model. They obtained that OSE-1 can reduce the deposition of Ab plaques and the neuroinflammation. Moreover, OSE-1 can ameliorate the cognitive impairment in murine model of Alzheimer. Despite the obtained results appear interesting and suitable for the use of OSE-1 in clinical trial for AD, the manuscript presents some Major Issues that need to be resolve to consent its publication.
1. The introduction needs to be improved because many concepts are dealt in an extremely concise manner, which make difficult to understand the background for those readers who are not expert in nutraceuticals and neurodegenerative disease. These include:
a. In the introduction and in abstract, some information about OSE-1 are lacking. What OSE-1 is? What does it acronym mean? How does it function as an antioxidant? What is the AD-related mechanism against which OSE-1 acts?
b. There is a need of a more accurate description of Alzheimer’s disease and its etiopathology to well understand the role and the function of OSE-1 (see below).
c. The aim of the study will be better explained, considering where the experiments will be performed and what the expected results are.
2. The section of “Material and Methods” requires a reorganization and an improvement.
a. The subsections would follow the timeline of the experiments.
i. Chemicals and Antibodies (including also the reagents used for extraction of OSE and U-HPLC)
ii. Plant Materials, Extract chemical isolation method and Ultra-high performance liquid chromatography (UHPLC) analysis
iii. Cell cultures
iv. Fluorescent measurement of intracellular reactive oxygen species (ROS)
v. Animals
vi. Neurocognitive assessment (Object recognition test, Radial Arm Maze Test)
vii. Tissue preparation
viii. Immunohistochemistry and Immunofluorescence
ix. Immunoblot
x. Statistical analyses
b. In the subsection 2.2 of the manuscript, it is reported the extraction of OSE-1 and OSE-2: please explain into Introduction the role of these two compounds and the differences.
c. The part relative to MTT assay and RT-PCR is lacking.
d. Statistical analyses will be improved because used post-hoc analyses (dunn’s method, tukey’s ….) are not specified.
3. In the results, some aspects are not clear:
a. Effects of OSE in cell viability of BV12 cells.
i. In “Material & Methods” section, the Authors isolated two OSE compound, i.e.,OSE-1 and OSE-2. It is unclear what of the two compounds was used for experiments.
ii. In Fig.1, the effects of OSE and AVN-1 on cells are reported. However, there is no evidences of AVN-1 in the text or in the legend of the Figure. It is better to explain if any difference with OSE is present in studied mechanism and to report into Introduction what it is and the related literature.
iii. In Fig. 1 A, it is not indicated how the treatment of OSE 200ug/uL significantly differ from other treatments.
b. OSE Exposure effectively inhibits brain A? expression.
i. It is better to talk of “Ab deposition and protein expression” rather than only “expression.”
ii. At the beginning of this subsection, the authors reported that obtained data are in according to ELISA assay. However, it is not clear if ELISA assay was performed contextually with these experiments (and in this case, this part will be added to the “material & methods” section and results will better explained) or are results of previously published manuscripts (and in this case, the Authors must add the reference).
iii. The Authors concluded the paragraph in this way : “The findings were in line with the Aβ findings.” However, no supporting reference is reported.
iv. Also in Figure 2, there is the presence of AVN-1, but its contribution is not reported in the text or in the legend of Figure.
c. BACE1 Expression in the Brain After OSE Exposure
i. No information about BACE1 and APP is reported in the introduction.
ii. To study BACE expression was used RT-PCR. However, no description of the procedure was found in section “Material and Methods”.
iii. Figure 3 displays results concerning AVN1, but no information about it was found in the text or in the legend of the figure.
d. OSE Exposure Decreases Neuroinflammatory Cells
i. General concepts about neuroinflammation in AD will be debated in the introduction to better understand the results that are summarized in this paragraph.
ii. Figure 4 reported results about AVN-1, but no explanation about it is reported in the text or in the legend of the figure.
e. Mechanism of OSE effects on AD-related phenotypes
i. The explanation about the linking between Ab pathway with other cellular mechanisms is not specified in the introduction. Although some concepts are then reported in the discussion, it is better to report them in the introduction to let the reader an exhaustive comprehension of the topic.
4. The Discussion section need to be improved considering :
a. Obtained results commented based on previous literature
b. The current state of art of treatments used in AD
c. If other studies about other nutraceuticals were performed, considering AD and other Neurodegenerative disorders
d. The potential of OSE-1 for the treatment of AD
e. Future perspectives
Moreover, the Authors must revise grammatically the manuscript because some sentences are difficult to understand and there are some mistakes in the punctuation.
Author Response
In this Manuscript, the Authors would investigate the effects of OSE-1, a natural compound, in ameliorating AD pathology, using an in vitro and an in vivo model. They obtained that OSE-1 can reduce the deposition of Ab plaques and the neuroinflammation. Moreover, OSE-1 can ameliorate the cognitive impairment in murine model of Alzheimer. Despite the obtained results appear interesting and suitable for the use of OSE-1 in clinical trial for AD, the manuscript presents some Major Issues that need to be resolve to consent its publication.
1. The introduction needs to be improved because many concepts are dealt in an extremely concise manner, which make difficult to understand the background for those readers who are not expert in nutraceuticals and neurodegenerative disease. These include:
(a.) In the introduction and in abstract, some information about OSE-1 are lacking. What OSE-1 is? What does it acronym mean? How does it function as an antioxidant? What is the AD-related mechanism against which OSE-1 acts?
Answer Thank you for your thoughtful comments to improve the manuscript. First, we have included oat seedlings extract (OSE) in the Abstract section. Second, we previously reported the OSE’s phytochemical profile as well as UPLC-CAD chromatograms of polyphenols in OSE [Changes in metabolites with harvest times of seedlings of various Korean oat (Avena sativa L.) cultivars and their neuraminidase inhibitory effects, Food Chemistry, 373(2022): 131429]. We have briefly added this information with this paper as a reference (19). OSE treatment affects JNK/ERK/p38 MAPK signaling, with considerable inhibition in p-JNK, p-p38, and p-ERK levels seen in the brain of OSE-treated Tg-5xFAD mice. Thus, OSE have gained recognition as neuroprotective agents in treating AD.
(b.) There is a need of a more accurate description of Alzheimer’s disease and its etiopathology to well understand the role and the function of OSE-1 (see below).
Answer We have added more detailed information regarding this point in the section of introduction.
Alzheimer's disease (AD) is one of the most prevalent neurodegenerative disorders and is attributed to several factors, including the emergence of amyloid-(A) plaques, hyperphosphorylation, microglia activation, and neuronal degeneration [12,13]. Progressive memory loss and cognitive impairment are common in AD patients. None of the available treatments for AD can delay or halt the progression of the disease; they can only treat symptoms. The abnormal accumulation of Aβ is still the key event in the onset and progression of AD, although there are many contributing factors [14-16]. The sequential cleavage of APP by β-secretases like BACE1 and γ-secretase complex produces Aβ. There are two main species, Aβ40 and Aβ42, with Aβ42 constituting the majority of Aβ plaques in the brain of AD patients [17–20]. Although postulated, the A mechanism of neurotoxicity has not yet been evidenced. Learning and memory deficits can be better understood by using an animal model of AD that exhibits Aβ pathology. The role of inflammation in AD is a more recent area of interest. Research utilizing animal models has shown that inflammation can impair cognitive function [21], as well as cause neuronal damage and synaptic loss in vivo and in vitro [22–24]. Although inflammation and microglia activation are thought to have a neuroprotective effect in short-term situations, over time they may cause neurotoxicity and an increase in Aβ load [21,25]. The activation of microglia by Aβ is thought to draw them to plaques and optimize phagocytosis [26]. Potentially, the microglial response to Aβ is secure, but after prolonged activation, the microglia take on a negative role and cause a degradation feed-forward loop [27].
(c.) The aim of the study will be better explained, considering where the experiments will be performed and what the expected results are.
Answer We have revised it regarding this point.
This study is unique in that it aimed to provide experimental evidence directly investigated how OSE1 affected AD by examining the cognitive function and exploring the inflammatory response’s mechanisms responsible for exerting these impacts.
2. The section of “Material and Methods” requires a reorganization and an improvement.
Answer We have revised it regarding this point.
(a.) The subsections would follow the timeline of the experiments.
Answer We have revised the order as follows.
(i.) Chemicals and Antibodies (including also the reagents used for extraction of OSE and U-HPLC)
(ii.) Plant Materials, Extract chemical isolation method and Ultra-high performance liquid chromatography (UHPLC) analysis
(iii.) Cell cultures
- Cell viability assay was added.
- Cell death detection assay was added.
(iv.) Fluorescent measurement of intracellular reactive oxygen species (ROS)
(v.) Animals
(vi.) Neurocognitive assessment (Object recognition test, Radial Arm Maze Test)
(vii.) Tissue preparation
(viii.) Immunohistochemistry
(ix.) Immunoblot
- qRT-PCR was added.
(x.) Statistical analyses
(b.) In the subsection 2.2 of the manuscript, it is reported the extraction of OSE-1 and OSE-2: please explain into Introduction the role of these two compounds and the differences.
Answer Thank you for your thoughtful comments to improve the manuscript. We are sorry, but there was a mistake in the process of writing the article. The difference between OSE (oat seedlings extract)-1 and OSE-2 is that the first extract is OSE-1, and the second extract is OSE-2. In this article, we all used the first extract, and the second extract, OSE-2, is meaningless. Therefore, OSE-2 was deleted and OSE-1 was all modified to OSE.
(c.) The part relative to MTT assay and RT-PCR is lacking.
Answer We have added more detailed information regarding this point in the section of M&M.
(d.) Statistical analyses will be improved because used post-hoc analyses (dunn’s method, tukey’s ….) are not specified.
Answer We have added more detailed information regarding this point in the section of M&M.
3. In the results, some aspects are not clear:
(a.) Effects of OSE in cell viability of BV12 cells.
(i.) In “Material & Methods” section, the Authors isolated two OSE compound, i.e.,OSE-1 and OSE-2. It is unclear what of the two compounds was used for experiments.
Answer The difference between OSE (oat seedlings extract)-1 and OSE-2 is that the first extract is OSE-1, and the second extract is OSE-2. In this article, we all used the first extract, and the second extract, OSE-2, is meaningless. Therefore, OSE-2 was deleted and OSE-1 was all modified to OSE. Thus, OSE-2 was deleted.
(ii.) In Fig.1, the effects of OSE and AVN-1 on cells are reported. However, there is no evidences of AVN-1 in the text or in the legend of the Figure. It is better to explain if any difference with OSE is present in studied mechanism and to report into Introduction what it is and the related literature.
Answer We are so sorry for our error and we used OSE and Avenacoside-type flavonoids (AVN-A) are the major compounds of OSE.
(iii.) In Fig. 1 A, it is not indicated how the treatment of OSE 200ug/uL significantly differ from other treatments.
Answer We have added the p-value in Figure.1 A
(b.) OSE Exposure effectively inhibits brain A? expression.
(I.) It is better to talk of “Ab deposition and protein expression” rather than only “expression.”
Answer We have revised it regarding this point.
(ii.) At the beginning of this subsection, the authors reported that obtained data are in according to ELISA assay. However, it is not clear if ELISA assay was performed contextually with these experiments (and in this case, this part will be added to the “material & methods” section and results will better explained) or are results of previously published manuscripts (and in this case, the Authors must add the reference).
Answer We have added more detailed information regarding this point.
(iii.) The Authors concluded the paragraph in this way : “The findings were in line with the Aβ findings.” However, no supporting reference is reported.
Answer We have added the reference regarding this point.
(iv.) Also in Figure 2, there is the presence of AVN-1, but its contribution is not reported in the text or in the legend of Figure.
Answer Avenacoside-type flavonoids (AVN-A) are the major compounds of OSE. We have added this description in the text or in the legend of Figure.
(c.) BACE1 Expression in the Brain After OSE Exposure
(i.) No information about BACE1 and APP is reported in the introduction.
Answer We have added more detailed information regarding this point in the section of introduction.
Alzheimer's disease (AD) is one of the most prevalent neurodegenerative disorders and is attributed to several factors, including the emergence of amyloid-(A) plaques, hyperphosphorylation, microglia activation, and neuronal degeneration [12,13]. Progressive memory loss and cognitive impairment are common in AD patients. None of the available treatments for AD can delay or halt the progression of the disease; they can only treat symptoms. The abnormal accumulation of Aβ is still the key event in the onset and progression of AD, although there are many contributing factors [14-16]. The sequential cleavage of APP by β-secretases like BACE1 and γ-secretase complex produces Aβ. There are two main species, Aβ40 and Aβ42, with Aβ42 constituting the majority of Aβ plaques in the brain of AD patients [17–20]. Although postulated, the A mechanism of neurotoxicity has not yet been evidenced. Learning and memory deficits can be better understood by using an animal model of AD that exhibits Aβ pathology. The role of inflammation in AD is a more recent area of interest. Research utilizing animal models has shown that inflammation can impair cognitive function [21], as well as cause neuronal damage and synaptic loss in vivo and in vitro [22–24]. Although inflammation and microglia activation are thought to have a neuroprotective effect in short-term situations, over time they may cause neurotoxicity and an increase in Aβ load [21,25]. The activation of microglia by Aβ is thought to draw them to plaques and optimize phagocytosis [26]. Potentially, the microglial response to Aβ is secure, but after prolonged activation, the microglia take on a negative role and cause a degradation feed-forward loop [27].
(ii.) To study BACE expression was used RT-PCR. However, no description of the procedure was found in section “Material and Methods”.
Answer We have added more detailed information regarding this point in the section of M&M.
(iii.) Figure 3 displays results concerning AVN1, but no information about it was found in the text or in the legend of the figure.
Answer Avenacoside-type flavonoids (AVN-A) are the major compounds of OSE. We have added this description in the text or in the legend of Figure.
(d.) OSE Exposure Decreases Neuroinflammatory Cells
(i.) General concepts about neuroinflammation in AD will be debated in the introduction to better understand the results that are summarized in this paragraph.
Answer We have added more detailed information regarding this point.
The role of inflammation in AD is a more recent area of interest. Research utilizing animal models has shown that inflammation can impair cognitive function [21], as well as cause neuronal damage and synaptic loss in vivo and in vitro [22–24]. Although inflammation and microglia activation are thought to have a neuroprotective effect in short-term situations, over time they may cause neurotoxicity and an increase in Aβ load [21,25]. The activation of microglia by Aβ is thought to draw them to plaques and optimize phagocytosis [26]. Potentially, the microglial response to Aβ is secure, but after prolonged activation, the microglia take on a negative role and cause a degradation feed-forward loop [27].
(ii.) Figure 4 reported results about AVN-1, but no explanation about it is reported in the text or in the legend of the figure.
Answer Avenacoside-type flavonoids (AVN-A) are the major compounds of OSE. We have added this description in the text or in the legend of Figure.
(e.) Mechanism of OSE effects on AD-related phenotypes
(i.) The explanation about the linking between Ab pathway with other cellular mechanisms is not specified in the introduction. Although some concepts are then reported in the discussion, it is better to report them in the introduction to let the reader an exhaustive comprehension of the topic.
Answer We have added more detailed information regarding this point in the section of introduction.
4. The Discussion section need to be improved considering :
(a.) Obtained results commented based on previous literature
Answer We have added more detailed information regarding this point in the section of Discussion.
We previously reported the OSE’s phytochemical profile as well as UPLC-CAD chromatograms of polyphenols in OSE (Food ChemistryVolume 373, Part B, 30 March 2022, 131429). Recent study reported the inhibitory effects of all compounds towards bacterial neuraminidase and thus, OSE may also have an important and useful resources for potential therapeutic or preventative agents against bacterial infection. To the best of our knowledge in the view point of AD.
(b.) The current state of art of treatments used in AD
Answer We have added more detailed information regarding this point in the section of Discussion.
Only symptoms of AD are currently being treated. Presently, the FDA has only approved four medications for AD: three AChE inhibitors (donepezil, galantamine, and rivastigmine), and one NMDA antagonist (memantine) [41]. Tacrine, a fifth drug, and another AChE inhibitor were taken off the market because it was hepatotoxic [42]. However, the benefits of these medications in treating the symptoms are only marginal. Additionally, they are ineffective in preventing the loss of neurons, brain atrophy, and ensuing gradual decline in cognition [41].
(c.) If other studies about other nutraceuticals were performed, considering AD and other Neurodegenerative disorders
Answer We have added more description regarding this point in the section of Discussion.
These results suggest that OSE may exhibit potential as a clinical treatment for neurodegenerative diseases, including Parkinson's disease, Alzheimer's disease, and senile dementia.
(d.) The potential of OSE-1 for the treatment of AD
Answer The overall goal of this study was to investigate whether OSE could provide protective effects against AD and, if positive, could be useful for future AD drug development.
(e.) Future perspectives
Answer We have added more description regarding this point in the section of Discussion.
The overall goal of this study was to investigate whether OSE could provide protective effects against AD and, if positive, could be useful for future AD drug development. And the results of this study are also expected to aid the development of suitable cultivars, which can be potential health functional foods.
Moreover, the Authors must revise grammatically the manuscript because some sentences are difficult to understand and there are some mistakes in the punctuation.
Answer We did English editing fully and revised all parts as your request.

Reviewer 2 Report
Review for nutrients-1913912
The manuscript by Eun Ho Kim et al., entitled “The Potential Neuroprotective Effects of Extracts from Oat Sprout against Alzheimer’s Disease” is the research report finding that treatment with OSE protects against Aβ25–35-induced cytotoxicity in BV12 cells. Tg-5xFAD AD mice treated with OSE showed a drastic reduction was observed in terms of numbers as well as the size of Aβ plaques within Tg-5xFAD mice expose to OSE. OSE’s neuroprotective impacts against neurodegeneration, synaptic dysfunction, and neuroinflammation induced by amyloid-beta. OSE does act as a neuroprotective agent to combat AD-specific apoptotic cell death, neuroinflammation, amyloid-beta accumulation, as well as synaptic dysfunctions in AD mice’s brains. Also, reported that OSE treatment affects JNK/ERK/p38 MAPK signaling, with considerable inhibition in p-JNK, p-p38, and p-ERK levels seen in the brain of OSEI-treated Tg-5xFAD mice. The study was well designed the results are confirmed with various techniques and the outcomes are significant. However, some remarks should be taken by authors under consideration before paper publication. The manuscript needs major revision before its final publication.
Comments:
1. Authors should thoroughly check the grammatical mistakes and spelling and space between the word throughout the manuscript.
2. Authors should include the expansion of OSE in the manuscript.
3. Authors should include the experimental timeline as a separate figure in the materials and methods section, it will better serve the readers.
4. Authors should explain what is OSE1 and OSE-2. Why did the authors choose OSE-1 for the treatment?
5. Authors must include the detailed experiment procedure and treatment procedure in the materials and methods section.
6. Authors should explain how did they administer OSE to the animals? How much dose the given to each animal for the treatment? How did they fix the dose for both animals and cell culture treatment?
7. Authors should include references for the behavior analysis procedures.
8. Authors should include the catalog number, dilution, and company name for each antibody used in the study.
9. In the materials and methods section, the authors mentioned Immunofluorescence (IF) methods used for GFAP analysis but in the results section, GFAP-represented images are looks like Immunohistochemistry (IHC) images. It should be clarified.
10. In IHC authors should include the IBA-1 antibody in the materials and methods section.
11. In the materials and method section, authors should include the detailed western blot procedure (like what primary and secondary antibodies were used and dilution, how the membrane was developed? how the western blot images were analyzed? and what software was used for the image analysis?)
12. Authors should provide higher magnification images for all the IHC and IF results, it will better serve the readers.
13. Authors should make sure the images are taken from the cerebral cortex “OR” the entorhinal cortex.
14. For the statistical analysis, the authors should provide a detailed procedure for the statistical methods used. (what software do they use for the analysis? Which post-hoc analysis do they use?)
15. In Fig. 4, the GFAP representation is not an IF image. The authors should explain.
16. In Fig. 4, the authors should include the graphical representation of the number of GFAP+ cells and IBA-1+ cell counting. It will strengthen the results.
17. For all the IF and IHC images authors should include the scale bar for the images.
18. Fig 5A should be represented as a separate figure in the materials and methods section.
19. In Fig 5, the authors should represent relative brain weight (g%).
20. Authors did the animal study and experiments, they mentioned in the materials and method section, “Experiments were carried out in compliance with approval from Institutional Animal Care and Use Committee. All institutional and international norms for animal care were complied with.” But in the Ethical Statement, they stated, “Our study did not require an ethical board approval because it did not contain human or animal trials”. Should explain.
Author Response
The manuscript by Eun Ho Kim et al., entitled “The Potential Neuroprotective Effects of Extracts from Oat Sprout against Alzheimer’s Disease” is the research report finding that treatment with OSE protects against Aβ25–35-induced cytotoxicity in BV12 cells. Tg-5xFAD AD mice treated with OSE showed a drastic reduction was observed in terms of numbers as well as the size of Aβ plaques within Tg-5xFAD mice expose to OSE. OSE’s neuroprotective impacts against neurodegeneration, synaptic dysfunction, and neuroinflammation induced by amyloid-beta. OSE does act as a neuroprotective agent to combat AD-specific apoptotic cell death, neuroinflammation, amyloid-beta accumulation, as well as synaptic dysfunctions in AD mice’s brains. Also, reported that OSE treatment affects JNK/ERK/p38 MAPK signaling, with considerable inhibition in p-JNK, p-p38, and p-ERK levels seen in the brain of OSEI-treated Tg-5xFAD mice. The study was well designed the results are confirmed with various techniques and the outcomes are significant. However, some remarks should be taken by authors under consideration before paper publication. The manuscript needs major revision before its final publication.
Comments:
1. Authors should thoroughly check the grammatical mistakes and spelling and space between the word throughout the manuscript.
Answer We did English editing fully and revised all parts as your request.
2. Authors should include the expansion of OSE in the manuscript.
Response: Thank you for your thoughtful comments to improve the manuscript. First, we have included oat seedlings extract (OSE) in the Abstract section. Second, we previously reported the OSE’s phytochemical profile as well as UPLC-CAD chromatograms of polyphenols in OSE [Changes in metabolites with harvest times of seedlings of various Korean oat (Avena sativa L.) cultivars and their neuraminidase inhibitory effects, Food Chemistry, 373(2022): 131429]. We have briefly added this information with this paper as a reference (19) in the Materials & Methods section.
3. Authors should include the experimental timeline as a separate figure in the materials and methods section, it will better serve the readers.
Answer We revised it per your comment.
4. Authors should explain what is OSE1 and OSE-2. Why did the authors choose OSE-1 for the treatment?
Answer Thank you for your thoughtful comments to improve the manuscript. We are sorry, but there was a mistake in the process of writing the article. The difference between OSE(oat seedlings extract)-1 and OSE-2 is that the first extract is OSE-1, and the second extract is OSE-2. In this article, we all used the first extract, and the second extract, OSE-2, is meaningless. Therefore, OSE-2 was deleted and OSE-1 was all modified to OSE.
5. Authors must include the detailed experiment procedure and treatment procedure in the materials and methods section.
Answer We have added it per your comment in the section of M&M.
5xFAD[B6SJL-Tg(APPSwFlLon,PSEN1*M146L*L286V)6799Vas/Mmjax]transgenic mice were purchased from the Jackson Laboratory (Bar Harbor, ME, USA). Female WT, as well as Tg-5xFAD mice, were used and allocated into various groups. Exposure to OSE1 began when the animals were 1.5 months old, which means that their brains were yet to mature. This is because the development of intraneuronal Aβ aggregation took place genetically within Tg-5xFAD from this point onward. Experiments were carried out in compliance with approval from Institutional Animal Care and Use Committee (DCIAFCR-220915-25-YR). All institutional and international norms for animal care were complied with. Following 1 week of adaption, all Tg-5xFAD mice were randomly divided into either the AD model group or the OSE + AD model group (n=5 in each group). C6 mice were also randomly divided into either the normal control (NC) group or the OSE + NC group (n=5 in each group). The mice in the OSE + NC and OSE + AD model groups were administered OSE (100 mg/kg) orally once daily for 6weeks. Following treatment, behavioral tests and biochemical experiments were performed.
6. Authors should explain how did they administer OSE to the animals? How much dose the given to each animal for the treatment? How did they fix the dose for both animals and cell culture treatment?
Answer We have added it per your comment in the section of M&M.
The mice in the OSE + NC and OSE + AD model groups were administered OSE (100 mg/kg) orally once daily for 6weeks. Following treatment, behavioral tests and biochemical experiments were performed.
7. Authors should include references for the behavior analysis procedures.
Answer We have added the reference per your comment.
8. Authors should include the catalog number, dilution, and company name for each antibody used in the study.
Answer We have added this information per your comment in the section of M&M.
9. In the materials and methods section, the authors mentioned Immunofluorescence (IF) methods used for GFAP analysis but in the results section, GFAP-represented images are looks like Immunohistochemistry (IHC) images. It should be clarified.
Answer We have revised our error and GFAP data is IHC not IF.
10. In IHC authors should include the IBA-1 antibody in the materials and methods section.
Answer We have added this information per your comment in the section of M&M.
11. In the materials and method section, authors should include the detailed western blot procedure (like what primary and secondary antibodies were used and dilution, how the membrane was developed? how the western blot images were analyzed? and what software was used for the image analysis?
Answer We have added this information per your comment in the section of M&M.
12. Authors should provide higher magnification images for all the IHC and IF results, it will better serve the readers.
Answer We have added this data per your comment in the section of result.
13. Authors should make sure the images are taken from the cerebral cortex “OR” the entorhinal cortex.
Answer We have clarified this point per your comment.
14. For the statistical analysis, the authors should provide a detailed procedure for the statistical methods used. (what software do they use for the analysis? Which post-hoc analysis do they use?)
Answer We have added this information per your comment in the section of M&M.
15. In Fig. 4, the GFAP representation is not an IF image. The authors should explain.
Answer We have revised our error and GFAP data is IHC not IF.
16. In Fig. 4, the authors should include the graphical representation of the number of GFAP+cells and IBA-1+ cell counting. It will strengthen the results.
Answer We have added this information per your comment in the section of result.
17. For all the IF and IHC images authors should include the scale bar for the images.
Answer We have added this information per your comment in the section of figure.
18. Fig 5A should be represented as a separate figure in the materials and methods section.
Answer We revised it per your comment.
19. In Fig 5, the authors should represent relative brain weight (g%).
Answer We have added this information per your comment in the section of result.
20. Authors did the animal study and experiments, they mentioned in the materials and method section, “Experiments were carried out in compliance with approval from Institutional Animal Care and Use Committee. All institutional and international norms for animal care were complied with.” But in the Ethical Statement, they stated, “Our study did not require an ethical board approval because it did not contain human or animal trials”. Should explain.
Answer We have removed our error.

Round 2
Reviewer 2 Report
The revised manuscript is well written and presented by the authors.
Author Response
Thank you for your help and recognition.